# Understanding Refeeding Syndrome in Critically Ill Patients: A Narrative Review

**DOI:** 10.3390/nu17111866

**Published:** 2025-05-29

**Authors:** Raffaele Borriello, Giorgio Esposto, Maria Elena Ainora, Giorgio Podagrosi, Giuliano Ferrone, Irene Mignini, Linda Galasso, Antonio Gasbarrini, Maria Assunta Zocco

**Affiliations:** 1CEMAD Digestive Diseases Center, Fondazione Policlinico Universitario “A. Gemelli” IRCCS, Università Cattolica del Sacro Cuore, Largo A. Gemelli 8, 00168 Rome, Italy; raffaele.borriello@unicatt.it (R.B.); giorgio.esposto@guest.policlinicogemelli.it (G.E.); mariaelena.ainora@policlinicogemelli.it (M.E.A.); irene.mignini@guest.policlinicogemelli.it (I.M.); linda.galasso@guest.policlinicogemelli.it (L.G.); antonio.gasbarrini@unicatt.it (A.G.); 2Department of Anesthesia and Intensive Care, Fondazione Policlinico Universitario “A. Gemelli” IRCCS, Università Cattolica del Sacro Cuore, Largo A. Gemelli 8, 00168 Rome, Italy; giorgio.podagrosi@guest.policlinicogemelli.it (G.P.); giuliano.ferrone@policlinicogemelli.it (G.F.)

**Keywords:** refeeding syndrome, refeeding, ICU, critical care, nutrition, hypophosphatemia

## Abstract

Refeeding syndrome (RS) is defined as the spectrum of metabolic and biochemical disorders related to rapid nutritional replenishment after a prolonged period of fasting. It is caused by an abrupt shift in electrolytes and fluid among intra- and extracellular compartments, leading to metabolic disturbances like hypophosphatemia, vitamin deficiency, and fluid overload. RS often remains underdiagnosed due to variability in definition and diagnostic criteria adopted, overlapping clinical features with other complications and low awareness among clinicians. Critically ill individuals, particularly those admitted to intensive care units (ICUs), represent a cohort with peculiar features that may heighten RS risk due to their baseline frailty, frequent undernutrition, and the metabolic stress of acute illness. However, studies specifically conducted in ICU settings have yielded conflicting results regarding incidence rates, prognostic impact, and specific risk factors. Despite these differences, all evidence consistently highlights RS as a frequent and serious complication in critically ill patients. Early detection and prevention are essential, relying on prompt nutritional assessment at ICU admission, careful monitoring of serum electrolytes before and during refeeding, and a conservative caloric approach to nutrient reintroduction, alongside supportive therapy and electrolyte supplementation if RS manifestations occur. Clinicians should be aware of the significant prevalence and potential severity of RS in critically ill patients, along with the ongoing challenges related to its early recognition, prevention, and optimal nutritional management. This review aims to provide a comprehensive overview of the current knowledge on the incidence, prognostic impact, risk factors, clinical manifestations, and nutritional management of RS in critically ill patients while highlighting existing evidence gaps and key areas requiring clinical attention.

## 1. Introduction

Refeeding syndrome (RS), which can be defined as the spectrum of the metabolic, hydroelectrolytic, and clinical alterations induced by an abrupt introduction of nutrients in a patient who spent a significant period with an absent or neglectable caloric intake [1], represents a potentially fatal complication of hospitalized patients, carrying a very significant impact in terms of morbidity and mortality [1,2,3]. Despite being originally described mainly in specific categories of individuals experiencing prolonged periods of starvation such as war prisoners [4] or subjects with anorexia nervosa [5], RS has increasingly been acknowledged during the last few decades as a frequent and often neglected complication detectable across a wide range of medical units, including internal medicine [6], gastroenterology [7], surgery [8], geriatrics [9], stroke units or neurologic facilities [10,11,12], onco-hematology [13], or even pediatric and neonatal units [14,15,16]. Critically ill patients or those admitted to intensive care units (ICUs) represent a peculiar category for whom this condition is of particular concern due to the presence of one or more organ failures, prolonged periods of hospitalization or starvation, and/or comorbidities influencing caloric intake [1]. Despite a significant prevalence of RS in intensive care units [2], this problem still remains challenging for intensive care clinicians due to the difficulty in early recognition, the lack of universally accepted diagnostic criteria, the low evidence regarding Its treatment or prevention, and the general complexity in the management of critically ill patients. The aim of this review is to offer a distinct and clinically oriented synthesis focused specifically on the latest issues regarding RS in the population of critically ill patients, where data are fragmented and often inconclusive, aiming to provide an overview of the current knowledge about RS incidence, impact, prevention, and early treatment in ICUs and exploring the future perspectives of nutritional support and realimentation in this setting.

## 2. Methods

This review is a narrative synthesis of the current evidence on refeeding syndrome, beginning with a general overview of its pathophysiology, clinical features, and diagnostic considerations and subsequently narrowing its focus to critically ill patients, specifically addressing the incidence, risk factors, prognostic impact, and management strategies in this population. A literature search was conducted in the PubMed/MEDLINE database up to March 2025. The following keywords and combinations were used: “refeeding syndrome”, “refeeding”, “critically ill”, “intensive care unit”, “ICU”, “risk factors”, “incidence”, “management”, “enteral nutrition”, “parenteral nutrition”, “hypophosphatemia”, and “prognosis”. References from relevant articles were also hand-searched to identify additional studies. Two independent reviewers screened the studies for eligibility. Articles were included if they provided data or expert consensus relevant to the epidemiology, clinical outcomes, diagnosis, prevention, or management of RS. For the ICU-specific focus, we included original research articles, reviews, consensus guidelines, and clinical trials that involved adult critically ill or ICU populations. Studies conducted exclusively on pediatric patients were excluded, as well as studies not available in the English language. A total of 203 articles were retrieved from the PubMed database. We screened the titles and abstracts of all studies for relevance, study population, and sample size. Then, we reviewed the full texts of the selected paper, including those providing meaningful information or significant data for the purpose of the present review. A total of 82 articles were eventually included.

## 3. Definition and Background

One of the key factors contributing to the heterogeneity of data in the literature concerning RS is the long-standing absence of a universally accepted definition, which has led to inconsistencies in its recognition, diagnosis, and reporting across different clinical settings [17]. RS can be defined as a series of metabolic disturbances related to the transition from a prolonged catabolic state to an anabolic state due to the abrupt reintroduction of nutrients [1]. This process triggers significant fluid and electrolyte shifts, leading to potentially severe hypophosphatemia, hypokalemia, hypomagnesemia, thiamine deficiency, peripheral edema, and serious clinical manifestations including neuromuscular dysfunction, cardiac failure, and respiratory insufficiency [1].

The first reports describing adverse events related to realimentation after prolonged starvation emerged after World War II [4,18]. This was particularly prominent in individuals who had experienced months of starvation due to war imprisonment or detention in concentration camps and undergoing a substantial reintroduction of caloric intake, and an unexpected rate of medical complications such as heart failure, respiratory failure, neurologic symptoms, and peripheral edema was observed [4,18]. The expression “refeeding syndrome” was proposed in a case series published in 1981 to describe the electrolytic and metabolic alterations—particularly severe hypophosphatemia—occurring in two chronically malnourished subjects undergoing aggressive total parenteral nutrition (PN) who died due to acute cardiopulmonary decompensation [19]. Subsequently, an increasing number of case reports and literature reviews identified refeeding syndrome as a concrete and frequent complication of inpatients with a particular incidence in specific conditions such as anorexia nervosa [5,20,21], alcohol use disorder [22,23], cancer [24,25] and, more generally, in frail, critically ill, or older patients [26,27,28]. Other authors also referred to this condition as “refeeding hypophosphatemia” (RH), as hypophosphatemia is the main biochemical disturbance of RS [29,30,31], to the point that numerous studies have even considered hypophosphatemia as the sole diagnostic criterion of RS [17].

In 2006, the National Institute for Health and Care Excellence (NICE) formally recognized the clinical significance of RS in its guidelines for adult nutrition support, establishing the first standardized criteria to identify patients at risk or high risk of developing RS [32]. The first consensus dedicated specifically to refeeding syndrome was published by the American Society for parenteral and enteral nutrition (ASPEN) in 2020 [1]. The ASPEN consensus proposed an operative definition of RS based on a measurable decline in at least one electrolyte among phosphorus, potassium, or magnesium or the manifestation of thiamine deficiency developing within five days of reintroducing or significantly increasing nutrient intake [1]; notably, they also categorized RS severity according to the percentage reduction in serum electrolytes or the presence of organ dysfunction [1]. However, despite the NICE and ASPEN definitions, the absence of a universally accepted diagnostic criterion for RS continues to pose challenges in clinical practice. This variability is reflected in significant differences in reported incidence rates, preventive strategies, and treatment approaches across studies, as highlighted by systematic reviews and meta-analyses [17,33].

## 4. Pathogenesis

During prolonged fasting, human metabolism adapts to the depletion of nutrients to preserve energy and maintain essential functions. After the depletion of hepatic glycogen stores (within 24 h), as starvation progresses, there is a metabolic shift from carbohydrate utilization towards the oxidation of fatty acids and lipid utilization to produce ketone bodies as primary energy sources while sparing glucose for glucose-dependent tissues [34,35]. In these conditions, there is a decline of insulin activity due to the reduced availability of carbohydrates, while the activity of counter-regulatory hormones (e.g., catecholamines and cortisol) become dominant [36]. As starvation progresses, there is a progressive depletion of intracellular substrates, comprising electrolytes and vitamins, due to a generalized contraction of the intracellular compartment [35]. However, this depletion does not exitate in significant alterations of serum electrolytes due to intracellular mechanisms of adaptation and the concomitant loss of total water [35,37].

In case of refeeding, the reintroduction of nutrients induces an abrupt surge in insulin secretion and, consequently, an immediate shift in metabolism from lipid oxidation back to glycolysis [1,38]. This phenomenon causes a sudden activation of intracellular enzymes, which requires cofactors and vitamins that are relatively deficient in the organism, causing an immediate increase in intracellular uptake and an abrupt drop of serum concentration [1,35]. This mechanism precipitates the hallmark electrolyte disturbances of RS.

In particular, hypophosphatemia occurs due to the increased demand of phosphorus for the biosynthesis of adenosine triphosphate (ATP) and 2,3-diphosphoglycerate [35]. Hypokalemia occurs due to the increased stimulation by insulin of the membrane transporter sodium–potassium ATPase, which induces a rapid influx of potassium into cells [39,40]. The mechanisms causing hypomagnesemia during RS are less understood, though magnesium is a critical cofactor for many intracellular enzymes [1,35]; notably, hypomagnesemia also exacerbates hypokalemia by increasing potassium wasting in the distal nephron, where magnesium-dependent enzymes regulate potassium excretion [41]. Thiamine deficiency occurs due to the sudden increase in glucose-dependent metabolic pathways, which place high demands on enzymes requiring thiamine as a cofactor [1]. Fluid overload and edema have also been associated with refeeding, even if the effective relationship with this syndrome is debated [1]. Indeed, even if hyperglycemia and hyperinsulinemia can induce sodium retention with a consequent extracellular fluid expansion [35], this mechanism alone seems insufficient to develop a significant fluid overload due to the partial osmotic balance given by potassium intracellular uptake [1]; furthermore, the term “heart failure” was originally used to also indicate sudden cardiac death or lethal arrhythmias in addition to congestive heart failure; thus, the interpretation of past case reports documenting this condition following refeeding may have been misinterpreted [1]. A subacute edema has also been described in specific patients, which could result from a capillary leak or the inactivation of natriuretic peptides due to hyperinsulinemia rather than direct sodium and water retention [1].

## 5. Clinical Manifestations

The clinical presentation of RS varies in severity and the organs affected, depending on the predominant electrolyte or vitamin deficiency.

Severe hypophosphatemia impairs muscle contractility due to the depletion of intracellular phosphate metabolites such as ATP; thus, its manifestation comprises muscle weakness, tetany, rhabdomyolysis, and acute respiratory failure due to respiratory muscle insufficiency; furthermore, myocardial insufficiency and arrhythmias may also occur, as phosphorus plays a role in the regulation of cardiac electrical activity [39,42]. Neurologic manifestations such as delirium or seizures are also possible [1]; additionally, phosphate depletion reduces 2,3-diphosphoglycerate (2,3-DPG) levels, increasing hemoglobin’s oxygen affinity and subsequently worsening peripheral tissue hypoxia [1,42].

Hypokalemia impairs the transmission of cardiac and nervous impulses and can lead to life-threatening arrhythmias such as ventricular fibrillation and torsades de pointes [42,43]. Other clinical manifestations include muscle weakness, hyporeflexia, respiratory depression, paralysis, and metabolic alkalosis [1,42].

Hypomagnesemia also encompasses neuromuscular disorders into its clinical manifestation, including tremors, fasciculation, tetany, or ataxia; notably, it may also induce neuropsychiatric symptoms or disorders like apathy or depression [42,44]. In addition, as already mentioned, hypomagnesemia may worsen concomitant hypokalemia and its related manifestation and can equally contribute to the genesis of cardiac arrhythmias [41,42,44]. Gastrointestinal manifestations like nausea or constipation may also occur both due to hypokalemia or hypomagnesemia [1].

Thiamine deficiency can manifest as Wernicke encephalopathy, characterized by ophthalmoplegia, ataxia, and confusion, which can progress to the irreversible Korsakoff psychosis if untreated, developing anterograde or retrograde amnesia and confabulation [42]. Another complication related to thiamine deficiency is beriberi, which results from impaired ATP production due to defective oxidative metabolism. Depending on the primary system affected, beriberi can manifest as either congestive heart failure with tachycardia, peripheral vasodilation, and edema (wet beriberi) or through a nervous impairment leading to peripheral neuropathy with muscle weakness, paresthesia, and hyporeflexia (dry beriberi) [1,45]. Finally, thiamin deficiency can also cause lactic acidosis due to a reduced conversion of lactate into pyruvate [42].

A detailed list of the clinical features derived from metabolic disturbances related to RS is shown in Table 1.

## 6. Diagnosis and Screening

The early recognition of RS represents one of the crucial aspects of RS management, but this step is significantly hindered by the lack of a universally accepted definition and the limited awareness of the condition among clinicians, leading to underdiagnosis and delayed treatment [2,46]. The primary risk factors identified for RS in the literature have included pre-existing malnutrition and specific comorbidities strongly associated with its development, such as anorexia nervosa, alcohol use disorder, and cancer. Consequently, the initial step in assessing RS risk relies on malnutrition screening tools and the recognition of these comorbidities [1,17,47].

The first dedicated screening criteria for identifying high-risk patients were introduced by NICE in 2006, incorporating parameters such as low body mass index (BMI), unintentional weight loss, minimal or absent caloric intake in the preceding days, pre-existent serum electrolyte abnormalities, and the presence of high-risk comorbidities [32]. Subsequent frameworks, including those proposed by Friedli et al. in 2018 [48] and the ASPEN consensus in 2020, proposed similar risk categories [1]. In particular, the ASPEN criteria stratified patients into moderate- or high-risk categories based on the degree of weight loss, BMI, fasting duration, the severity of pre-existing electrolyte imbalances, the presence of high-risk comorbidities, and the entity of muscle or subcutaneous fat loss [1].

The diagnosis of refeeding syndrome is also challenging for the already mentioned inconsistencies in its definition and the absence of globally accepted diagnostic criteria, as well as the variability in its clinical manifestations. This condition is often recognized retrospectively, as the onset of the symptoms usually occurs only after the mentioned biochemical shifts and often when complications have already developed [17,49]. Therefore, RS diagnosis requires a high index of suspicion and an early identification of patients at risk. A wide range of diagnostic criteria have been proposed in the literature, ranging from the only appearance of hypophosphatemia after nutrient replenishment to broader definitions including the presence of at least one severe electrolyte disturbance and/or clinical signs of fluid overload or organ dysfunction after refeeding [17]. The differences among diagnostic criteria are the main cause of the variability among epidemiological data and confusing definitions; indeed, while hypophosphatemia is the only universally recognized feature of this condition, several authors diagnosed RS based only on serum biochemical abnormalities, while others considered the presence of clinical manifestations to be necessary [2].

Friedli et al. integrated NICE guidelines adding the diagnostic criteria for “imminent” or “manifest” RS according to the reduction in phosphate or any other two electrolytes and the presence of clinical manifestations related to these abnormalities [48]. Subsequently, refined criteria were proposed by the King’s College Hospital of London, requiring both electrolyte shifts and the concomitant presence of peripheral edema and organ failure [50]. More recently, the ASPEN consensus introduced diagnostic criteria which not only determine the presence of RS but also classify its severity based either on the degree of serum electrolyte reduction (phosphate, magnesium, and potassium) or clinical manifestation. According to ASPEN criteria, a diagnosis of mild RS can be made when electrolyte levels decrease by 10–20% from baseline after refeeding and a moderate RS occurs when the reduction exceeds 30%, while severe RS occurs when signs or symptoms of dyselectrolytemia or thiamine deficiency are present, regardless of serum levels [1]. All of the mentioned conditions must occur within 5 days from refeeding or a substantial increase in caloric intake [1] (Table 2).

## 7. Refeeding Syndrome in Critically Ill Patients: General Considerations and Incidence

Critically ill patients, particularly those admitted to intensive care units, represent a unique cohort at risk for refeeding syndrome due to the presence of one or more pre-existing organ failures and other stress-related disturbances induced by critical illness, making this population particularly at risk of serious complications [51]. Indeed, the stress of critical illness triggers a hypercatabolic state driven both by proinflammatory mediators, such as tumor necrosis factor (TNF) and interleukin-1 (IL-1), and a concomitant hormonal imbalance characterized by elevated levels of cortisol, glucagon, and catecholamines. This cascade promotes protein catabolism and impairs the ability of tissues to effectively utilize circulating nutrients, ultimately leading to lean mass wasting and significant weight loss [52]. Notably, this process seems to be refractory to artificial nutritional supplementation, at least during its acute phase [52]. Furthermore, critically ill patients or those admitted to ICUs often present with pre-existent malnutrition [53] and experience long periods of fasting due to invasive procedures or hemodynamic instability, further increasing their risk of developing RS [1].

Moreover, RS manifestations may also mimic other conditions which usually affect patients in ICUs, such as sepsis or fluid overload [1,51], complicating its differential diagnosis and thus requiring a high degree of clinical suspicion and awareness of this condition.

The exact incidence rate of RS in patients hospitalized in intensive care units is unknown, but the majority of studies suggest that it is a frequent and clinically significant complication in this setting. The first systematic review on RS, conducted by Friedli in 2017, evidenced an incidence rate of RS in ICUs of 34–52% [17]. Similar data were reported by an updated systematic review and meta-analysis published in 2021, which reported an incidence of 17–52%, also evidencing a higher incidence of RS in ICUs than in ordinary medical facilities [33]. The reason for this variability lies in the different definitions used to diagnose RS, being more prevalent in cases where the authors considered the presence of clinical manifestations unnecessary to make a diagnosis [33].

A large retrospective study published in 2023 that focused on 2123 ICU patients confirmed the high variability in RS incidence due to differing diagnostic criteria, evidencing an incidence rate ranging from 1.5% to even 88% according to eight different definitions [54].

The prospective study conducted in surgical ICU patients by Buitendag et al. also confirmed the same variability, observing that RS incidence was only 1.5% when using King’s College criteria, rising to 12.5% when adopting ASPEN criteria [8].

When restricted to either Friedli definition or the ASPEN criteria, a recent retrospective analysis in medical ICU patients reported an incidence of 22.7–27.3%, respectively [55]. Beyond diagnostic inconsistencies, another key factor contributing to this variability is the heterogeneity of ICU subspecialties and populations. Patients admitted to ICUs may need different kinds of medical support (e.g., mechanical ventilation, artificial nutrition, or cardiocirculatory assistance) and present with varying degrees of organ dysfunction or disability and different grades of RS risk. Indeed, studies conducted in specialized ICUs or specific cohorts of critically ill patients reported significantly different incidence rates [28,56,57].

A summary of the main studies reporting RS incidence in critically ill patients, comprising those analyzed in the two systematic reviews mentioned, is provided in Table 3.

## 8. Prognostic Impact of RS in Critically Ill Patients

Due to the severity of its manifestations, refeeding syndrome (RS) has long been associated with life-threatening outcomes and mortality since its earliest descriptions [4,19,47]. However, its actual prognostic impact in ICU patients remains unclear. Several studies have reported conflicting results, with significant variability depending on the definition of RS used and the specific ICU subpopulations examined.

A prospective study by Marik et al. (1996) involving surgical and medical ICU patients found a significant association between RH and prolonged mechanical ventilation and in-hospital length of stay (LOS), although mortality data were not reported [29]. Similarly, a 2014 retrospective study by Coşkun et al. observed an increased LOS and higher mortality in ICU patients with RH [58]. Conversely, several other studies have failed to demonstrate a significant association between RS and poor outcomes. A 2018 prospective study by Md Ralib in a Malaysian ICU reported no significant differences in ICU or hospital LOS or mortality between patients with and without RH [59]. At least three retrospective studies also found no correlation between RS and adverse outcomes, including one by Suzuki in 2013 involving 2730 general ICU patients [61], another by Fuentes (2016) in a surgical ICU population [56], and a third by Olthof involving mechanically ventilated patients [28]. These findings were further confirmed by a recent prospective study [60]. Also, the retrospective study by Tongyoo, which recognized RS using both NICE and ASPEN diagnostic criteria, found no significant difference in 30-day mortality between medical ICU patients with and without RS [55]. However, the prognostic impact of RS may be variable according to specific ICU subgroups. For example, in neurocritically ill patients, a retrospective study showed that RH was associated with a longer stay in the neurocritical care unit, higher 30-day and 6-month mortality, and poorer functional outcomes at 6 months [57].

Thus, despite the consistently reported incidence of RS, its relationship with clinical outcomes remains uncertain. Discrepancies across studies likely originate not only from differing definitions of RS, often including the use of different phosphate cut-offs and from different subspecialistic ICU cohorts considered, but also from variations in the grade of severity of patients analyzed [28,59].

Overall, RS may represent a serious complication in many critically ill patients, with a potentially greater impact in specific subgroups characterized by specific comorbidities or high baseline clinical complexity.

## 9. Risk Factors in Critically Ill Patients

The identification of RS risk in critically ill patients may be very challenging. The conventional RS screening criteria proposed by NICE and ASPEN were not based on the specific characteristics of critically ill patients and include conditions already prevalent in most ICU patients, such as prolonged fasting or weight loss [1,53]. While the ASPEN consensus suggests that all critically ill patients should be considered at intrinsic risk of RS when reintroducing calories [1], NICE guidelines do not make specific recommendations for this category. Their application in the ICU setting has shown that nearly half of the patients may be considered at high or very high risk for RS [62], but despite a potential prognostic utility [62,63], NICE criteria have demonstrated poor sensitivity in this setting [8]. Thus, several authors have highlighted the unmet need for specific evidence-based tools to assess RS risk and guide its management in critically ill patients [64,65]. Unfortunately, in this field of knowledge, studies aiming to identify predictive factors for RS in ICU populations have yielded uncertain results. Established risk factors recognized in general wards, such as low BMI and baseline low serum electrolyte levels prior to refeeding, have shown predictive value in critically ill patients as well [60,66,67]. Some studies have evidenced a possible predictive role of low serum prealbumin in this setting [29,59,67], while other authors evidenced an association among the administration of high doses of insulin or diuretics in ICUs with RS [8,68]. However, current evidence remains insufficient or inconsistent to support specific recommendations. Notably, even widely accepted factors in other settings, such as the duration of fasting before refeeding, have shown uncertain results in the ICU context [66]. Several authors have also explored the association between RS risk and scores of clinical severity such as APACHE II, the Sequential Organ Failure Assessment (SOFA) score, and the Glasgow Coma Scale (GCS), but also in this case, findings remain inconclusive or limited to specific ICU subgroups [29,55,59,66,69,70,71]. For instance, in neurocritically ill patients, combining GCS with ASPEN criteria may improve RS recognition [71]. Due to the lack of ICU-specific screening instruments, current recommendations emphasize a prompt assessment of nutritional status and regular serum electrolyte monitoring prior and after refeeding, combined with careful clinical observation and individualized management to prevent RS-related complications [65,72,73,74].

## 10. Avoidance of RS and Nutritional Management of Refeeding in ICUs

The avoidance and the prevention of RS worsening largely overlap, since there is no etiologic therapy for RS and the management of this condition relies on early detection, supportive measures, and a cautious management of refeeding [1]. Early detection should begin, as mentioned, with a comprehensive nutritional assessment at admission to the ICU including the evaluation of serum electrolytes, particularly phosphate levels [74]. In patients identified as being at risk of RS, serum potassium, magnesium, and phosphorus levels should also be measured prior to refeeding. In cases of moderate-to-high risk, the ASPEN consensus suggests considering a delay in the initiation of nutritional support until electrolyte abnormalities have been corrected, with a concomitant supplementation of thiamine 100 mg before refeeding or before the administration of dextrose-containing fluids [1]. Thiamine supplementation should be continued up to 5–7 days in case of high-risk comorbidities, alongside multivitamin supplements [1]. Despite being originally associated with parenteral refeeding [19], current evidence indicates that RS can develop following any form of nutritional reintroduction, including oral diet, enteral nutrition (EN), PN, or intravenous dextrose. Therefore, no specific route of nutrient delivery is currently recommended for RS prevention [1]. Interestingly, a study by Zeki et al. reported a higher incidence of RS with EN compared to PN, possibly due to an increased stimulation of GLP-1 and incretin release with enteral feeding, leading to higher insulin levels [75]. While earlier studies suggested better clinical outcomes with EN compared to PN in critically ill patients, more recent evidence does not clearly support a significant advantage of one modality over the other in this setting, even if European guidelines still support EN over PN as the preferred route [74,76].

In patients diagnosed with RS, energy intake should be significantly reduced. The only randomized multicenter clinical trial in critically ill patients with RS demonstrated improved 60-day and overall survival rates in those receiving caloric restriction (20 kcal/h for 48 h) compared to standard caloric intake [27,74]. These findings were also supported by the retrospective study by Olthof et al., which also found improved survival associated with caloric restriction in RS patients [28], confirming caloric restriction as the mainstay in RS treatment.

The approach to feeding initiation in patients at risk of RS represents another discussed and crucial aspect in RS management. Indeed, both the ASPEN consensus and NICE guidelines recommend a very cautious approach in nutrient reintroduction in order to lower RS risk, but the evidence regarding the effective benefit of restrictive refeeding protocols is weak, both for the general population and critically ill patients [77]. The ASPEN consensus suggests initiating refeeding with 100–150 g of dextrose or 10–20 kcal/kg on the first day, alongside intensive clinical and biochemical monitoring and with a gradual progression to full caloric goals [1], while the NICE guidelines advocate for an even more conservative strategy, recommending a starting intake of no more than 10 kcal/kg/day or as low as 5 kcal/kg/day in patients with particularly high risk [32]. These extremely cautious indications have been recently discussed by several authors, as studies adopting both restrictive and permissive caloric approaches in refeeding have actually shown a similar incidence rate of RS [77,78], while excessive restrictive protocols may heighten the risk of malnutrition-related complications. Also, in this case, evidence is actually conflicting. A recent study in patients requiring PN observed that higher amounts of energy at refeeding are associated with a higher risk of RS and that the caloric amount provided on the first day through PN is an independent predictor of RS [79]. Discrepancies among studies may also probably depend in this case on clinical differences among studied cohorts and RS definitions, with the consequent impossibility of giving strong recommendations for the best refeeding protocol in patients at risk [77,78].

Interestingly, in addition to the importance of the caloric regimen, a potential key factor influencing the risk of RS may also be represented by the qualitative composition of the macronutrients delivered. A prospective cohort study in critically ill patients with coronavirus disease 2019 (COVID-19) reported that a higher protein intake during the refeeding phase was associated with a lower incidence of RS, suggesting a potential benefit of protein supplementation in this population [80]. Conversely, a retrospective study in mechanically ventilated patients with established RS found that high protein nutritional support was associated with increased six-month mortality, although no association was observed with short-term outcomes. The authors hypothesized the existence of a time- and dose-dependent effect of protein intake on RS risk during specific phases of refeeding in critically ill patients [81]. These findings underscore the need for further research into the role of macronutrient composition in RS prevention and outcomes.

Given the complexity of RS prevention and management in this population, the early involvement of clinical nutrition specialists can be pivotal in guiding individualized nutritional strategies. Moreover, a multidisciplinary approach in the intensive care setting has been associated with improved clinical outcomes [82]. An overview of the key aspects and uncertain areas contributing to RS management in critically ill patients is provided in Figure 1.

## 11. Conclusions

Refeeding syndrome is increasingly recognized as a challenging and insidious complication in hospitalized patients; however, specific tools for its assessment in distinct populations, such as the critically ill, are still lacking. Clinicians should be aware of the heightened vulnerability of ICU patients, whose frailty and clinical complexity may amplify both the risk and severity of RS. In the absence of targeted therapy, prevention and early detection remain essential. A comprehensive nutritional assessment at ICU admission should be conducted alongside careful monitoring of electrolyte levels at baseline, prior to, and during refeeding. A conservative approach to nutrient reintroduction is recommended, together with thiamine and electrolyte supplementation, supportive therapy, and caloric restriction in the case of RS-related symptoms or complications. Further studies are needed to clarify several aspects, including the identification of ICU subgroups at highest risk of RS-related morbidity, the most relevant risk factors specific to this population, optimal refeeding protocols following starvation, and the role of macronutrient composition in the pathogenesis of RS. Given the multifactorial nature of RS and the challenges in its recognition, the early involvement of clinical nutrition specialists and a multidisciplinary team approach is highly recommended to improve patient safety and clinical outcomes.

## Figures and Tables

**Figure 1 nutrients-17-01866-f001:**
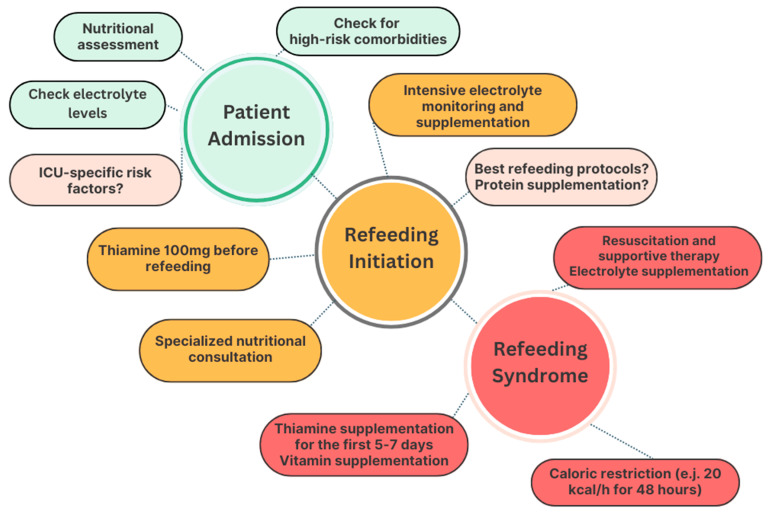
Overview of clinical phases and uncertainties in the management of refeeding syndrome in critically ill patients. The recognition and management of RS in this setting can be conceptually divided into three key stages, namely patient admission to the ICU, reintroduction of nutrients, and manifest refeeding syndrome. Around each phase, relevant clinical practices (green/orange/red bubbles) and areas of ongoing uncertainty or debate (pale pink bubbles) are illustrated. While the importance of early nutritional assessment, RS risk stratification, and electrolyte monitoring and supplementation is well established, further research is needed to better define ICU-specific risk factors and to determine optimal refeeding protocols in terms of both caloric progression and macronutrient composition.

**Table 1 nutrients-17-01866-t001:** Clinical features of refeeding syndrome.

Hypophosphatemia	Hypokalemia	Hypomagnesemia	Thiamine Deficiency	Fluid Overload
muscle weakness tetany paresthesias rhabdomyolysis delirium seizures acute respiratory failure acute heart failure arrhythmias peripheral tissue hypoxia hypotension shock	muscle weakness arrhythmias torsade de pointes hyporeflexia muscle paralysis respiratory depression metabolic alkalosis nausea vomiting constipation	muscle weakness tremors fasciculation tetany ataxia seizures apathy depression nausea vomiting constipation	Wernicke encephalopathy Korsakoff psychosis dry beriberi wet beriberi lactic acidosis	acute heart failure pulmonary edema peripheral edema

**Table 2 nutrients-17-01866-t002:** Summary of the main RS definitions proposed.

RS Definition	Diagnostic Criteria
Refeeding hypophosphatemia	Various degrees of % phosphate reduction after refeeding or drop below predefined cut-offs [17]
NICE guidelines integrated by Friedli et al. [32,48]	Imminent RS: Decrease in phosphate from baseline >30% or <0.6 mmol/L OR any two other electrolytes decrease below normal range within 72 h after start of nutrition therapy without clinical manifestations Manifest RS: Clinical symptoms of RS associated with the previous phosphate or electrolyte shift within 72 h after start of nutrition therapy
King’s College criteria [50]	Severely low electrolyte concentrations (potassium below <2.5 mmol/L, phosphate < 0.32 mmol/L, or magnesium < 0.5 mmol/L) associated with peripheral edema or acute circulatory fluid overload and organ dysfunction including respiratory failure, cardiac failure, and pulmonary oedema
ASPEN consensus criteria [1]	Mild RS: A decrease in any one, two, or three of the following: serum phosphorus, potassium, and/or magnesium levels by 10–20% within 5 days of reinitiating or substantially increasing energy provision Moderate RS: A decrease in any one, two, or three of the following: serum phosphorus, potassium, and/or magnesium levels by 20–30% within 5 days of reinitiating or substantially increasing energy provision Severe RS: A decrease in any one, two, or three of the following: serum phosphorus, potassium, and/or magnesium levels by >30% and/or organ dysfunction resulting from a decrease in any of these and/or due to thiamine deficiency within 5 days of reinitiating or substantially increasing energy provision

**Table 3 nutrients-17-01866-t003:** Incidence of RS among critically ill patients across various clinical studies.

RS Definition	Incidence Rate (%)	Study Design	Population	No. of Patients	Author	Year
Refeeding hypophosphatemia	34%	Prospective study	Surgical and medical ICU patients	62	Marik et al. [29]	1996
Refeeding hypophosphatemia	52.1%	Retrospective study	Medical ICU patients	117	Coşkun et al. [58]	2014
Refeeding hypophosphatemia	39%	Retrospective study	Surgical ICU patients	213	Fuentes et al. [56]	2017
Refeeding hypophosphatemia	36.8%	Retrospective study	Mechanically ventilated ICU patients	337	Olthof et al. [28]	2018
Refeeding hypophosphatemia	42.6%	Prospective study	Surgical and medical ICU patients	109	Md Ralib et al. [59]	2018
Refeeding hypophosphatemia	17.1%	Retrospective study	Neurocritically ill patients	328	Xiong et al. [57]	2020
King’s College criteria—ASPEN criteria	1.5–12.5%	Prospective study	Surgical ICU patients	200	Buitendag et al. [8]	2021
Eight different definitions including refeeding hypophosphatemia and ASPEN criteria	from 1.5% to 88%	Retrospective study	Medical ICU patients	2123	Naik et al. [54]	2023
Refeeding hypophosphatemia	47.2%	Prospective study	Medical ICU patients	195	Schneeweiss-Gleixner et al. [60]	2024
NICE criteria and ASPEN criteria	22.7–27.3%	Retrospective study	Medical ICU patients	216	Tongyoo et al. [55]	2025

## Data Availability

No new data were created or analyzed in this study. Data sharing is not applicable to this article.

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
