# Peer review of "Understanding Refeeding Syndrome in Critically Ill Patients: A Narrative Review"

_nutrients, 2025, doi:10.3390/nu17111866_

Round 1
Reviewer 1 Report
Comments and Suggestions for Authors
Congratulations on your significant achievement.
Regarding the literature search, I have a few minor requests for clarification. Please specify the number of papers retrieved from the databases for analysis and provide a flowchart illustrating your literature selection and processing methodology.
Author Response
We thank you for your appreciation and your suggestions. We integrated the methods paragraph highlighting the keywords and method of literature search, the number of articles retrieved and the article selection strategy.
Reviewer 2 Report
Comments and Suggestions for Authors
I have read and evaluated the article entitled Understanding Refeeding Syndrome in critically ill patients: a narrative review.
The article is very well written, and the division and arrangement of the chapters make the text much easier to read. The topic addressed is very important for practitioners because of the serious consequences and the lack of clear guidelines for diagnosis.
The article contains all the required elements that form a logical whole.
I regret that the authors did not try to create a systematic review, which would have significantly improved the data quality. However, given that this is a narrative review, I still suggest that the methods section of the article would have been beneficial to the readability of the entire article if it had included the number of researchers analyzing the articles, the number of peer-reviewed articles that contained the search keywords, the number of eligible articles, and the handling of discussion papers.
For better understanding, I would modify Table 3 so that the definition of RS and other information about RS are repositioned on the left side, and the authors and year on the right side. This is just a cosmetic editorial suggestion. It would be better to focus on RS rather than the article.
Author Response
Thank you for your comment and suggestion. We provided the information required and we summarized the article selection strategy in the methods section. We also modified the table according to your comment.Reviewer 3 Report
Comments and Suggestions for Authors
I thank the authors for giving me the opportunity to read this interesting review and congratulate them on their work.
Author Response
We really appreciate your comment. Thanks.